# LoRA-Switch: Boosting the Efficiency of Dynamic LLM Adapters via System-Algorithm Co-design

## Abstract

Recent literature has found that an effective method to customize or further improve large language models (LLMs) is to add dynamic adapters, such as low-rank adapters (LoRA) with Mixture-of-Experts (MoE) structures. Though such dynamic adapters incur modest computational complexity, they surprisingly lead to huge inference latency overhead, slowing down the decoding speed by 2.5+ times. In this paper, we analyze the fine-grained costs of the dynamic adapters and find that the fragmented CUDA kernel calls are the root cause. Therefore, we propose LoRA-Switch, a system-algorithm co-designed architecture for efficient dynamic adapters. Unlike most existing dynamic structures that adopt layer-wise or block-wise dynamic routing, LoRA-Switch introduces a token-wise routing mechanism. It switches the LoRA adapters and weights for each token and merges them into the backbone for inference. For efficiency, this switching is implemented with an optimized CUDA kernel, which fuses the merging operations for all LoRA adapters at once. Based on experiments with popular open-source LLMs on common benchmarks, our approach has demonstrated similar accuracy improvement as existing dynamic adapters, while reducing the decoding latency by more than 2.4 times.

## 1 Introduction

Large language models (LLMs) have demonstrated remarkable capabilities in language understanding and generation. To customize the pretrained models to vertical domains or further enhance their capabilities, various adapter techniques such as Low-Rank Adapters (LoRA) (Hu et al., 2021), LLaMA-Adapter (Zhang et al., 2023), and Prompt Tuning (Lester et al., 2021) have been employed with great success. These methods are acclaimed for boosting the accuracy of LLM without extensive training, thus facilitating efficient model customization and performance enhancement. Among these, dynamic adapters (Feng et al., 2024; Gou et al., 2024; Liu et al., 2023a; Luo et al., 2024) represent an even more potent strategy to augment the capacity of adapters. By integrating conditionally computed lightweight adapters into the pretrained model, dynamic adapters allow for selective fine-tuning of adapter parameters. This technique not only maintains the original strengths of the model but also substantially increases its adaptability and capacity.

However, we found that despite the relatively minor impact of dynamic adapters on parameter size and computing complexity (typically adding only 1-5% of the origin model), they may introduce significant latency overhead. For instance, the dynamic adapters that we studied all increase decoding inference latency by 250-950%. The seemingly modest computational complexity of the low-rank matrices employed results in substantial extra CUDA kernel execution latency, surpassing that of models without dynamic adapters. This dramatic increase in latency is primarily attributed to the prolonged execution time of context operations during CUDA kernel runs, which considerably exceeds the actual computation time. Dynamic adapters often require four or more additional CUDA kernel calls for each layer, in stark contrast to just a single call needed for the forward computation of the original backbone matrix. This excessive number of context operations substantially amplifies the latency overhead, leading to a severe escalation of inference latency.

Reducing the inference latency overhead of dynamic adapters is challenging. Existing dynamic adapters (Dou et al., 2024; Feng et al., 2024; Gao et al., 2024; Gou et al., 2024; Li et al., 2024; Liu et al., 2023a; Luo et al., 2024; Wu et al., 2024a; Yang et al., 2024) adopt block-wise or layer-wise routing structures, where activated LoRA adapters must be computed separately. If we were to pre-merge activated LoRA adapters into the backbone weights for forward computation, akin to the strategy employed by LoRA (Hu et al., 2021), it would fundamentally reduce the number of CUDA kernel calls. However, this merging changes the parameters of the LLM model, and for the next input, different adapters might be activated. Thus, after processing the current input, it becomes necessary to unmerge the activated adapters from the LLM model that was altered. The routing dynamicity involved in such merging processes becomes prohibitively costly. For instance, the widely adopted dynamic adapter structure, the Mixture of Experts (MoE), determines the activated adapters based on the output of the last layer. Each layer of dynamic adapters must then wait for its router to compute the gating scores before it can proceed to merge the activated LoRA adapters. This fragmented operational approach can inadvertently introduce even greater overhead costs.

Our approach to addressing the challenge is based on a holistic system-algorithm co-design. Specifically, we have developed a MoE-based dynamic adapters structure that facilitates token-wise adapter routing. Each token is associated with $k$ weighted paths of LoRA adapters, activated prior to the decoding of the token. This setup ensures that, although the model is enhanced with dynamic structures, the inference process for each token remains relatively static due to the pre-determined adapters. To further enhance the efficiency, we pre-merge the activated LoRA adapters into the pretrained model's backbone before each token's decoding. This strategy fundamentally reduces the CUDA kernel execution overhead, thereby significantly lowering latency. With this innovative setup, we have re-engineered the inference process to seamlessly switch and merge adapters for each token, aligning the process closely with the original pretrained LLM's token decoding. Another pivotal component of our system is the development of a fused CUDA kernel, named SGMM, which efficiently manages the activated and inactivated adapters. This engineering solution ensures a smooth integration of dynamic adapters, optimizing both performance and efficiency.

We evaluate our LoRA-Switch design across a range of benchmarks, comparing it against multiple state-of-the-art dynamic adapter baselines. The experiment results demonstrate that our approach are comparable with well-established strong baselines. Notably, our method significantly reduces the running overhead associated with other dynamic adapter alternatives, achieving an average speedup of 2.4 times in decoding latency.

In summary, our contributions are as follows:

- We uncover the high latency overhead introduced by dynamic adapters, which is a practical issue usually neglected by existing approaches. We analyze the fundamental reasons behind such high overhead, providing insights on the computational bottlenecks.

- We introduce a novel architecture for dynamic adapters, named LoRA-Switch. This design enhances the capacity of LLM adapters while minimizing the latency overhead, thereby offering an optimal balance between performance and efficiency.

- Through extensive experiments, we demonstrate that LoRA-Switch not only achieves accuracy on par with existing dynamic adapters across a variety of general and domain-specific tasks, but it also cuts down decoding inference latency by more than 2.4 times.

## 2 BACKGROUND AND MOTIVATION

### 2.1 DYNAMIC ADAPTERS

Given the strengths of both the Mixture of Experts (MoE) (Jiang et al., 2024; Shazeer et al., [n. d.]; Snowflake AI Research Team, 2024; The Mosaic Research Team, 2024; xAI, 2024) and Low-Rank Adaptation (LoRA) (Hu et al., 2021), their integration has become a focal point of recent research efforts. Recent studies (Feng et al., 2024; Gao et al., 2024; Gou et al., 2024; Liu et al., 2023a; Luo et al., 2024) have explored combining these two techniques to further augment the capabilities of large language models (LLMs). This integration leverages the scalability of MoE and the efficiency of LoRA, proposing a promising pathway to meet the escalating demands for model performance and

efficiency. Formally, the computation process of dynamic adapters can be formulated as Equation 1:

$$y^l = f^l(x^l) + \sum_{i=1}^{N} G^l(x^l)_i E_i^l(x^l), \tag{1}$$

where the superscript $l$ means $l$-th layer, $N$ represents number of adapters experts, $G^l(x^l) =$ Softmax(TopK($W_g^l x^l$)) represents the top-k (typically top-2) router in the dynamic adapters block, $f^l$ represents the pretrained backbone in $l$-th layer, and $E^l(x^l) = W_{up}^l(W_{down}^l(x^l))$ represents the output of LoRA experts, where the matrix $W_g^l$ and $W_{up}^l/W_{down}^l$ are the trainable parameter matrix of the router network and LoRA experts, respectively.

## 2.2 Unexpected Latency Overhead of Dynamic Adapters

Table 1: Inference cost of different dynamic adapters.

| Method | Decoding latency (ms/token) | Parameter size (B) | FLOPS (G) |
|---|---|---|---|
| Llama2-7B | 2.4 | 6.74 | 6.61 |
| MOLA (Gao et al., 2024) | 25.3 (+954%) | 7.07(+4.89%) | 6.65(+0.61%) |
| PESC (Wu et al., 2024b) | 8.5 (+254%) | 6.97(+3.41%) | 6.64(+0.45%) |
| MoRAL (Yang et al., 2024) | 8.6 (+258%) | 6.97(+3.41%) | 6.67(+0.91%) |

Although dynamic adapters can enhance accuracy and involve only a modest increase in parameter size and computing complexity, they unfortunately introduce a substantial inference latency overhead. We evaluate different dynamic adapters methods with Llama2-7B (Touvron et al., 2023a) on ShareGPT (OpenChat, 2023) dataset for 50 queries one by one, and generate 200 new tokens for each query. We report the decoding latency of processing the 50 queries. As demonstrated in Table 1, existing methods involving dynamic adapters result in an approximate 1%-5% increase in parameter count and less than a 1% increase in computing complexity measured in FLOPS. However, these enhancements lead to a substantial increase in decoding latency, with overheads ranging from 200% to 950%.

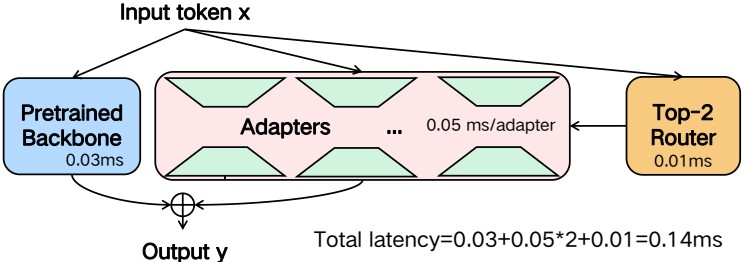

Figure 1: Decoding phase execution time profiling of one dynamic adapters layer in MoRAL (Yang et al., 2024). Note: The execution time results were preceded by a warm-up of 100 executions and are obtained on the average of 300 executions.

To elucidate the sources of latency overhead introduced by dynamic adapters, we conducted a granular analysis of latency within various components during the decoding phase. As illustrated in Figure 1, it is evident that the execution time for the adapters (0.05ms) exceeds that of the pretrained backbone (0.03ms). Despite the relatively modest computational complexity of the LoRA adapters employed in dynamic configurations, each adapter necessitates dual launches of CUDA kernel context operations. The execution time of these CUDA kernels does not correlate linearly with the size of the matrices involved, leading to considerable latency in the adapter components. This, in turn, significantly escalates the overall inference latency, highlighting a critical area for optimization in dynamic adapter architectures. More profiling results can be found in Appendix A.

## 2.3 CHALLENGE OF REDUCING LATENCY OVERHEAD OF DYNAMIC ADAPTERS

A straightforward way to reduce inference latency overhead is to reduce the times of CUDA kernel context operations. Like LoRA (Hu et al., 2021), one couple pre-merge adapters into the original matrix and then perform token decoding computation. We implement this simple strategy in MoRAL by directly merging activated adapters layer by layer before computing, although it decreases the number of CUDA kernel launches. However, the additional operations introduced higher latency, where the decoding latency is 4.5 ms/token, which is still 88% higher than the original LLM model. This is because merging a LoRA adapter into the backbone matrix requires an additional invocation of a CUDA kernel to perform the matrix multiplication for the up and down projections. Due to these inherent computational complexities, optimizing the existing structure of dynamic adapters further is challenging.

To effectively leverage the structure of dynamic adapters, we introduce LoRA-Switch. Unlike the existing layer-wise or block-wise dynamic routing mechanisms, our approach LoRA-Switch utilizes token-wise routing strategy, which fundamentally reduces the number of CUDA kernel calls, thereby reducing decoding inference latency.

## 3 DESIGN OF LoRA-SWITCH

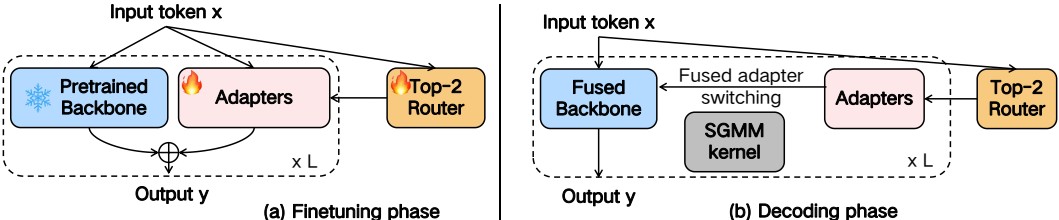

Figure 2: Overview of LoRA-Switch.

## 3.1 OVERVIEW

As shown in Figure 2, given a pre-trained LLM, we first extend it with LoRA-Switch to enhance its model capacity and then finetune it on either general or domain-specific datasets. During the decoding phase, each token is initially processed through a router to compute the gating for each layer. Then we implemented an SGMM kernel that facilitates fast fused adapter switching. This advanced functionality enables the rapid merging of activated adapters into the backbone and the efficient unmerging of inactivated adapters from the backbone, significantly reducing the number of CUDA kernel calls. This process ensures that the decoding speeds are comparable to those of a origin pretrained model, optimizing performance without compromising efficiency.

## 3.2 MODEL STRUCTURE

In LoRA-Switch, we extend adapters only in the linear layers of the pre-trained backbone, and we insert a Top-2 router $G^1$ only at the first expanded linear layer.

**Finetuning phase.** As shown in Figure 2 (a), during fine-tuning, the computation process of LoRA-Switch can be written as Equation 2:

$$y^l = f^l(x^l) + \sum_{i=1}^{N} G^1(x^1)_i E_i^l(x^l), \tag{2}$$

where LoRA-Switch only replaces the $G^l(x^l)$ with $G^1(x^1)$ in Equation 1, where $x^1$ denotes the input in the first expanded linear layer. LoRA-Switch employs a token-wise pre-gated LoRA structure, meaning that the routing weights for all layer adapters are identical. This design not only preserves the model's dynamicity but also facilitates latency optimization during inference.

**Decoding phase.** As depicted in Figure 2 (b), LoRA-Switch leverages the token-wise pre-gated structure to reduce latency during the decoding phase. During the decoding phase of LLM generation, where the input $x$ is a single token, the Top-2 router $G^1$ determines which experts' adapters are activated. Then the activated experts will be merged into pretrained backbone. Finally, The fused backbone perform forward process and the decoding computation can be written as Equation 3:

$$y^l = f_*^l(x^l) \tag{3}$$

To efficiently calculate $f_*^l$ for all layers, we propose to perform fused adapter switching (Section 3.3) with the SGMM (Section 3.4) kernel, which merge the parameters of all activated experts adapters across all layers into the original parameters of pretrained model in a single CUDA kernel operation. Finally, the fused backbone execute just like the initial pretrained backbone as shown in Equation 3. This streamlined integration significantly enhances the efficiency of the decoding process.

**Prefilling phase.** The latency of LoRA-Switch during the prefilling phase is comparable to that of existing dynamic adapters. For the prefilling phase, we have not implemented specific optimizations, as the latency in the LLM generation stage primarily originates from the decoding phase.

### 3.3 FUSED ADAPTER SWITCHING

To calculate $f_*^l = f^l + \sum_{i=1}^{N} G^1(x^1)_i E_i^l$ according to Equation 1 and Equation 2, we may merge multiple experts adapters into backbone because one input token may activate multiple adapters (typically Top-2). The experts in LoRA-Switch are LoRA adapters, which contain down projection LoRA_DOWN and up projection LoRA_UP. Then we can compute $f_*^l$ as Euquation 4:

$$f_*^l = f^l + \sum_{i=1}^{N} G^1(x^1)_i \cdot (\text{LoRA\_DOWN}_i^l \times \text{LoRA\_UP}_i^l), \tag{4}$$

We only need to invoke the CUDA kernel $k$ times instead of $N$ times, as $G^1(x^1)$ specifically targets the top-$k$ selections. To further reduce the number of CUDA kernel calls, we concatenate all LoRA adapters before merging them into the original model parameters as Equation 5:

$$\text{LoRA\_DOWN}^l = concat([G^1(x^1)_i \cdot \text{LoRA\_DOWN}_i^l, \ i = 1, ..., N]),$$
$$\text{LoRA\_UP}^l = concat([\text{LoRA\_UP}_i^l, \ i = 1, ..., N]). \tag{5}$$

Thus, we can merge the concatenated LoRA adapters into origin backbone as Equation 6:

$$f_*^l = f^l + \text{LoRA\_DOWN}^l \times \text{LoRA\_UP}^l. \tag{6}$$

Although Equation 4, Equation 5, and Equation 6 enable a reduction in CUDA kernel calls, they necessitate the storage of $f^l$ to compute $f_*^l$, which approximately doubles the GPU memory overhead. We observe that during the decoding phase, the activated adapters varying from one token to the next. A simple way to obtain $f^l$ is to unmerge the activated adapters by the last token in the current iteration as Equation 7

$$f^l = (f_*^l)^{t-1} - (\text{LoRA\_DOWN}^l)^{t-1} \times (\text{LoRA\_UP}^l)^{t-1}, \tag{7}$$

where the superscript $(t-1)$ denotes the operations from the last input iteration. Then Equation 6 and Equation 7 can be rewritten as Equation 8

$$(f_*^l)^t = (f_*^l)^{t-1} - (\text{LoRA\_DOWN}^l)^{t-1} \times (\text{LoRA\_UP}^l)^{t-1}$$
$$+ (\text{LoRA\_DOWN}^l)^t \times (\text{LoRA\_UP}^l)^t. \tag{8}$$

Then we concatenate the activated adapters of current input and inactivated adapters of last input as Equation 9:

$$\text{Fused\_LoRA\_DOWN}^l = concat([-(\text{LoRA\_DOWN}^l)^{t-1}, (\text{LoRA\_DOWN}^l)^t]),$$
$$\text{Fused\_LoRA\_UP}^l = concat([-(\text{LoRA\_UP}^l)^{t-1}, (\text{LoRA\_UP}^l)^t]). \tag{9}$$

So the concatenated LoRA adapter switching operation can be rewritten from Equation 8 to Equation 10:

$$(f_*^l)^t = (f_*^l)^{t-1} + \text{Fused\_LoRA\_DOWN}^l \times \text{Fused\_LoRA\_UP}^l. \tag{10}$$

To calculate Equation 10, we utilize our efficiently designed CUDA kernel, SGMM, to seamlessly integrate these fused adapters into the pretrained LLM backbone for all layers with only one CUDA kernel call.

Finally, the whole decoding process of LoRA-Switch are shown in Algorithm 1.

---

**Algorithm 1** The token decoding process of LoRA-Switch.

---

**Inputs:** $x$: input token at time $t$, and all model parameters.
**Outputs:** Logits prediction of the next token.
 1: Calculate $G^1(x^1)$ as Equation 2.
 2: Concatenate current activated adapters as Equation 5.
 3: Concatenate activated and inactivated adapters as Equation 9.
 4: Perform fused adapter switching with SGMM as Equation 10.
 5: Execute model forward as Equation 3 and obtain next token logits prediction.
 6: Return logits prediction.

---

### 3.4   SGMM KERNEL

The straight-forward way to merge the LoRA adapter into the backbone is to merge them layer by layer. This approach requires multiple calls to the CUDA kernel, which introduces additional latency due to kernel launches. Moreover, smaller kernel computations underutilize GPU thread blocks, leading to a low GPU throughput. We observe the layer-by-layer merging operations can be handled concurrently and introduce a CUDA kernel called Segmented Gather Matrix Multiplication (SGMM) to finally handle the merging of LoRA adapters of LoRA-Switch, adapted from the concept of SGMV proposed by Punica (Chen et al., 2023).

SGMM is designed to execute a batched GEMM operations, which can be summarized by the following Equation 11:

$$f_* = f + \text{Fused\_LoRA\_DOWN} \times \text{Fused\_LoRA\_UP}, \tag{11}$$

where $f_*$ is the resultant updated matrix of the backbone; $f$ is the original weight matrix of the backbone; Fused_LoRA_DOWN and Fused_LoRA_UP are the adapter matrices for weight matrix $f$. The addition operation within the SGMM kernel is performed in place, significantly reducing the additional memory overhead. This optimization ensures that memory usage is minimized, enhancing the overall computational efficiency of our system.

When wrapping these operations into a single CUDA kernel, taking full advantage of the GPU's computational resources is challenging. Each thread block should be fully utilized, and the computational load needs to be balanced across them. To achieve this, we divide the matrix multiplication into multiple GEMM tiles and assign them to different thread blocks. On the other hand, these thread blocks must switch context with global memory/shared memory frequently, thus causing significant latency. To tackle this, we adopt a pre-fetch buffer mechanism to hide loading latency. It fetches data from higher-level memory before the next matrix operation to boost memory access efficiency.

The input parameters for SGMM are arrays of pointers to the LoRA matrices and the backbone matrices, which respectively store the corresponding entries of each layer's LoRA matrix and the shape of each matrix segment. When launching the kernel, SGMM applies as many thread blocks as possible and divides the large matrix multiplication into multiple GEMM tiles of the same shape, with each tile operating matrix computation. Based on the input pointers and shapes of the original matrices, SGMM calculates the address and size of each tile in global memory and assigns each of them to a thread block. The optimal tiling scheme related to hardware is selected to ensure the full utilization of each thread block. To hide the loading latency of tile switching, SGMM uses two buffers in shared memory and CUDA core registers, respectively; one for the current tile's matrix computation and the other for pre-fetching the next tile's data into the buffer. This design stems from the same shape of sequential tiles, which makes us able to predict the next tile to be processed. In our methodology, thread blocks are executed concurrently, facilitating the simultaneous processing required for our computational tasks. This parallel execution enables the efficient merging and unmerging of LoRA weights, a critical operation in our approach.

# 4 EVALUATION

In this section, we present the experimental results to show the superiority of the proposed LoRA-Switch in accuracy and runtime efficiency performance.

## 4.1 EXPERIMENT SETUP

**Datasets/benchmarks.** To demonstrate the proposed LoRA-Switch could augment general ability of LLM, we follow PESC (Wu et al., 2024b) and simultaneously fine-tuned the model on a diverse set of skills, including encompassing coding, mathematical, and other general abilities from various subjects. This training involved integrating three distinct datasets from varied domains during the instruction tuning phase: SlimORCA (Lian et al., 2023), Magicoder (Wei et al., 2023), and MetaMathQA (Yu et al., 2023) datasets. We utilize LM-Eval-Harness (Gao et al., 2023) as tool to evaluate general ability on ARC (Clark et al., 2018), HellaSwag (Zellers et al., 2019), MMLU (Hendrycks et al., 2021), TruthfulQA (Lin et al., 2022), WinoGrande (Sakaguchi et al., 2019), and MT-Bench (Zheng et al., 2023) benchmarks and report the accuracy.

Also, to demonstrate the proposed LoRA-Switch could improve domain specific ability of LLM, we follow MoLA (Gao et al., 2024) and fine-tuned the model on downstream task. We evaluate three recent question-answering benchmarks, including ScienceQA(Lu et al., 2022), CommonsenseQA(Talmor et al., 2019), and OpenbookQA(Mihaylov et al., 2018). We follow the task-specific fine-tuning framework to evaluate their effectiveness.

To evaluate runtime efficiency performance of LoRA-Switch, we utilize real-world sharegpt (Open-Chat, 2023) dataset to simulate user queries, which is used in many LLM serving frameworks (Cai et al., 2024; Kwon et al., 2023; Miao et al., 2024). We serve 50 queries from sharegpt dataset one by one, and generate 200 new tokens for each query. Finally, we report the inference time of processing the 50 queries.

**Baselines.** We compare LoRA-Switch with four PEFT approaches, including LoRA (Hu et al., 2021), layer-wise gating dynamic adapters like MoRAL (Yang et al., 2024) and MOLA (Gao et al., 2024), and block-wise gating dynamic adapters PESC (Wu et al., 2024b). We also compare full-parameter fine-tuning. Due to the extensive fine-tuning requirements of both the MoLA and Full-Parameter methods, which necessitate over a week of training in general LLM tasks, we find this duration to be impractical. Consequently, we have opted not to include these methods in the comparisons within general task comparison. We use LLama2-7B (Touvron et al., 2023b) and Mistral-7B (Jiang et al., 2023) as the pretrained base LLM, which is the default base model for most dynamic adapters.

**Implementation Details.** For general tasks, we set the number of experts as 8 and LoRA rank as 64 for all dynamic adapters methods, and LoRA alpha is set to 16 and LoRA dropout is 0.05, following the default LoRA settings.. The models underwent instruction tuning for one epoch with about 12 hours. We use a constant learning rate schedule with a warm-up ratio of 0.03, and the paged AdamW (Dettmers et al., 2023; Loshchilov and Hutter, 2019) optimizer with a learning rate of 2e-4, no weight decay, a batch size of 256, and a sequence length of 512 tokens. For domain specific tasks, the number of experts is 8 and the rank of each LoRA expert is also 8, and we adopt top-2 for the router. LoRA alpha is set to 16 and LoRA dropout is 0.05, following the default LoRA settings. we trained 20 epochs for downstream task fine-tuning about 6 hours. We use AdamW (Loshchilov and Hutter, 2019) as the optimizer with a learning rate of 3e-4. The cutoff length is set to 256 and the batch size is 128. For LoRA baseline, We applied it to four weight matrices in the self-attention module ($W_q$, $W_k$, $W_v$, $W_o$) and three weight matrices in the MLP module ($W_{gate}$, $W_{down}$, $W_{up}$). All the fine-tuning tasks were conducted on the servers with eight A100-80GiB GPUs. For runtime performance evaluation, we use one A100-80GiB GPU as serving server.

## 4.2 ACCURACY COMPARISON

### 4.2.1 IMPROVING GENERAL CAPABILITY OF LLM

As depicted in Table 2, our approach considerably enhances the ability of LLMs to perform on general tasks. On average, our method achieves an accuracy of 60.12%, which is an improvement of approximately 0.5% over traditional LoRA fine-tuning techniques. Furthermore, our method's performance is highly competitive with existing dynamic adapters, trailing the average accuracy

of the MoRAL method by a mere 0.1%. The accuracy of all dynamic adapter methods not only surpasses that of the standard LoRA approach but is also significantly higher than that of the un-tuned Llama2-7B model, affirming the effectiveness of dynamic adapters in boosting LLM performance.

Moreover, detailed evaluation on individual datasets reveals that our method surpasses LoRA on the ARC, HellaSwag, MMLU, TruthfulQA, and Winogrande datasets. Notably, when compared to other dynamic adapters, LoRA-Switch achieves the highest accuracy on the MMLU and TruthfulQA datasets, highlighting its substantial potential and robust performance.

Table 2: Accuracy of incremental training achieved with different dynamic adapters.

| Method | ARC | HellaSwag | MMLU | TruthfulQA | Winogrande | Avg |
|---|---|---|---|---|---|---|
| Llama2-7B (base) | 51.71 | **77.74** | 48.30 | 45.31 | 72.45 | 59.10 |
| LoRA | 51.79 | 77.02 | 50.46 | 45.13 | 73.80 | 59.64 |
| MoRAL (layer-wise) | 52.13 | 77.57 | 51.10 | 45.93 | **74.35** | 60.22 |
| PESC (block-wise) | **53.58** | 77.27 | 51.07 | 46.04 | 74.27 | **60.45** |
| LoRA-Switch (ours) | 52.39 | 77.60 | **51.15** | **46.15** | 73.32 | 60.12 |

Table 3: Accuracy of incremental training achieved with different dynamic adapters on MT Bench

| Methods | MT Bench score |
|---|---|
| Llama2-7B (base) | 6.07 |
| LoRA | 6.13 |
| LoRA-Switch | 6.25 |
| MoRAL | 6.26 |
| PESC | 6.28 |

Table 4: Accuracy of domain-specific fine-tuning achieved with different dynamic adapters on Llama2-7B.

| Methods | ScienceQA | CommonsenseQA | OpenbookQA | Avg |
|---|---|---|---|---|
| Llama2-7B (base) | 53.19 | 47.82 | 45.80 | 48.94 |
| Full-Parameter | 93.12 | 77.48 | 80.40 | 83.67 |
| LoRA | 91.01 | 75.51 | 77.00 | 81.17 |
| MoLA (layer-wise) | **91.91** | 77.89 | **82.80** | **84.20** |
| MoRAL (layer-wise) | 90.74 | 76.41 | 76.60 | 81.25 |
| PESC (block-wise) | 90.02 | 76.00 | 78.40 | 81.47 |
| LoRA-Switch (ours) | 91.39 | **79.03** | 80.40 | 83.60 |

### 4.2.2 DOMAIN-SPECIFIC CUSTOMIZATION

Table 4 and Table 5 illustrates that our method markedly enhances LLM performance on specific downstream tasks. Achieving an average accuracy of 83.58% on Llama2-7B, our approach shows a notable improvement of 2.41% over traditional LoRA fine-tuning. Additionally, the performance of our method remains highly competitive with other dynamic adapters, only slightly trailing the average accuracy of the MoLA method by 0.62%.

This demonstrates the dynamic adapters' ability to significantly outperform both the standard LoRA method and the un-tuned Llama2-7B model, thereby verifying their effectiveness in refining the performance of large language models. Our method also demonstrates resilience by achieving an accuracy marginally lower by only 0.09% compared to Full-Parameter fine-tuning, further underlining the efficacy of our proposed approach.

Further analysis on individual datasets shows our method outperforming LoRA on the ScienceQA, CommonsenseQA, and OpenbookQA datasets. Our approach also achieves the highest accuracy on

Table 5: Accuracy of domain-specific fine-tuning achieved with different dynamic adapters on Mistral-7B.

| Methods | ScienceQA | CommonsenseQA | OpenbookQA | Avg |
|---|---|---|---|---|
| Mistral-7B (base) | 62.24 | 58.93 | 57.8 | 59.66 |
| LoRA | 94.15 | 79.85 | 84.2 | 86.06 |
| MoRAL (layer-wise) | 93.79 | 81.57 | 85.8 | 87.05 |
| PESC (block-wise) | 94.33 | 80.46 | 86.4 | 87.06 |
| LoRA-Switch (ours) | 93.82 | 81.29 | 86.6 | **87.24** |

the CommonsenseQA dataset when compared with other dynamic adapters, emphasizing both the substantial potential and the robust performance of LoRA-Switch.

These findings solidify the effectiveness of our proposed dynamic adapter strategy, making a compelling case for its adoption in enhancing the capabilities of LLMs across varied downstream tasks and domains.

## 4.3 RUNTIME PERFORMANCE

As reported in Table 6, we have evaluated the inference latency and peak GPU memory usage of our method and various baseline methods on the ShareGPT dataset in response to user queries. The results show that our method exhibits significantly lower decoding latency compared to all other dynamic adapter methods, being 2.7 times faster than the previously fastest method, PESC. Furthermore, our method's decoding latency is less than 30% higher than that of the original Llama2-7B model.

Overall, all dynamic adapter methods introduce additional decoding latency, ranging from 250% to 900%. MOLA exhibits the highest latency due to the incorporation of dynamic adapters at every linear layer within the model, resulting in a substantial number of required CUDA GEMM operations. In contrast, methods like PESC and MoRAL, which only add dynamic adapters to each MLP layer, demonstrate considerably lower decoding latency compared to MOLA.

Table 6: Latency and memory overhead of different dynamic adapters.

| Method | Decoding latency (ms/token) | Peak Memory (GiB) |
|---|---|---|
| Llama2-7B | 2.4 | 12.9 |
| MOLA (layer-wise) | 25.3 (+954%) | 26.3 (+104%) |
| PESC (block-wise) | 8.5 (+254%) | 13.1 (+2%) |
| MoRAL (layer-wise) | 8.6 (+258%) | 13.3 (+3%) |
| LoRA-Switch (ours) | 3.1 (+29%) | 13.8 (+7%) |

Regarding peak GPU memory usage, the implementations of PESC, MoRAL, and LoRA-Switch exhibit only a modest increase of 2%-7% compared to the original Llama2-7B model. This indicates that our newly proposed dynamic adapter structures have minimal requirements for GPU memory. In contrast, MOLA exhibits significantly higher GPU memory consumption, which can be attributed to the extensive number of dynamic adapters it incorporates internally. This difference highlights the efficiency of our approach in managing hardware resources while maintaining performance.

## 4.4 ABLATION STUDY

In our initial experiments, we rigorously assess the influence of diverse adapter configurations on the fine-tuning accuracy of our system, henceforth referred to as LoRA-Switch. The empirical results consistently demonstrate that LoRA-Switch significantly outperforms the baseline Llama2-7B model across a variety of adapter setups such as adjustments of the LoRA adapter rank, the number of adapter experts, and Top-K routing.

Further investigation is conducted into the individual contributions of the components within LoRA-Switch. Notably, the experimental findings reveal a substantial increase in latency when the fused adapter switching mechanism is omitted, underscoring the efficacy of our proposed approach.

More details can be found in Appendix B.

## 5 RELATED WORK

**Parameter efficient fine-tuning (PEFT).** PEFT stands out as the optimal approach for fine-tuning pretrained large language models (LLMs). Among the myriad of popular techniques are Adapters (Houlsby et al., 2019), Prefix Tuning (Li and Liang, 2021), Prompt Tuning (Lester et al., 2021), P-tuning (Liu et al., 2023b), and LoRA (Hu et al., 2021), each offering unique advantages. This paper specifically explores the efficacy of LoRA, which has demonstrated superior performance and have the most efficient inference performance.

**Dynamic adapters.** Existing dynamic adapters can be categorized based on their gating strategies into block-wise and layer-wise types, where adapters are added either to every block or to every layer, respectively. Block-wise methods such as MOLA (Gao et al., 2024), MoELoRA (Luo et al., 2024), MoCLE (Gou et al., 2024), and MOELoRA (Liu et al., 2023a) integrate adapter branches within a single Transformer block. Conversely, layer-wise approaches like MoRAL (Yang et al., 2024), PESC (Wu et al., 2024b), and LoRAMoE (Dou et al., 2024) incorporate adapters within the MLP's linear layers. While these methods introduce a minimal amount of parameters, they typically result in high inference latency. In contrast, our work introduces LoRA-Switch, a novel token-wise dynamic adapter approach that effectively integrates with system optimizations to significantly reduce inference latency. This innovation represents a substantial improvement over traditional dynamic adapter configurations.

## 6 CONCLUSION

This paper addresses the significant challenge of inference latency in dynamic adapters for large language models (LLMs). Through a comprehensive analysis, we have identified the underlying factors contributing to increase latency in traditional block-wise and layer-wise adapter models. Our proposed solution, LoRA-Switch, introduces a novel token-wise dynamic adapter configuration that leverages system-level optimizations to dramatically reduce inference latency without compromising the model's adaptability and performance. The effectiveness of LoRA-Switch has been validated through rigorous experiments, demonstrating substantial improvements in efficiency across both general and domain-specific datasets. These findings not only enhance our understanding of dynamic adapter architectures but also set a new benchmark for future research in efficient LLM tuning.

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

## A  PROFILING LATENCY OVERHEAD OF DYNAMIC ADAPTERS

To meticulously investigate how the inference latency of LoRA adapters and the pretrained backbone varies with increasing computational demands, we conducted tests across different input sequence lengths and examined the relationship between adapter rank and inference latency.

As shown in Figure 3, regardless of whether it is during the prefilling or decoding phase, and irrespective of the LoRA ranks being high or low, the latency of the LoRA adapters consistently exceeds that of the backbone. This phenomenon is primarily attributed to the number of CUDA kernel calls rather than the computing complexity involved in each call. The underlying reason is that the latency associated with CUDA kernel calls does not scale linearly with computing complexity. This insight highlights a crucial aspect of system behavior that significantly impacts the performance of dynamic adapters.

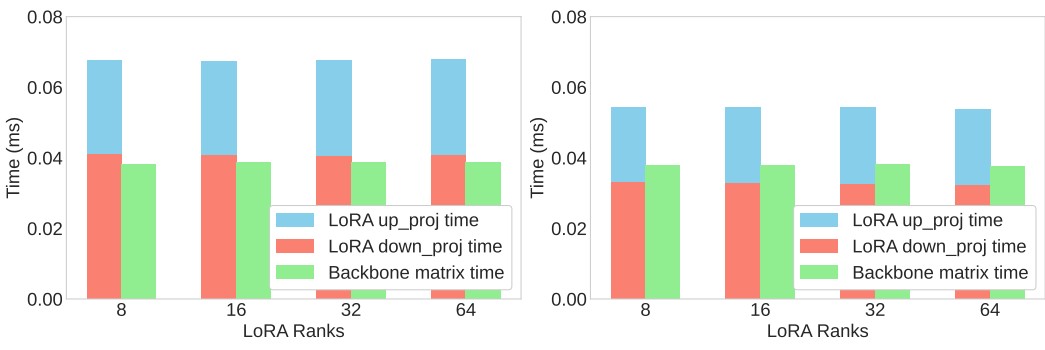

(a) Prefilling phase: sequence length is 100.    (b) Decoding phase: sequence length is 1.

Figure 3: Latency breakdown of one dynamic adapter layer under different settings.

# B ABLATION STUDY EXTENSION

## B.1 ABLATION STUDY ON MODEL PERFORMANCE

Firstly, we explore the impact of various adapter configurations on the fine-tuned accuracy of LoRA-Switch. We experimented with four modified configurations: a) Setting the number of experts per layer to 16; b) Setting the rank of each LoRA expert to 16; c) Setting the rank of each LoRA expert to 32; d) Changing the Top-2 routing to Top-1 routing. Experiments were conducted on both general and domain-specific tasks.

Table 7: Ablation study on general LLM ability improvements of LoRA-Switch.

| Method | ARC | HellaSwag | MMLU | TruthfulQA | Winogrande | Avg |
|---|---|---|---|---|---|---|
| LLama2-7B | 51.71 | 77.74 | 48.30 | 45.31 | 72.45 | 59.10 |
| LoRA-Switch (16 expert) | 51.71 | 77.34 | 50.42 | 45.33 | 73.95 | 59.75 |
| LoRA-Switch ($r$=32) | 51.11 | 77.16 | 50.30 | 44.63 | 73.01 | 59.24 |
| LoRA-Switch ($r$=16) | 51.28 | 76.91 | 49.82 | 43.95 | 74.11 | 59.21 |
| LoRA-Switch (Top-1) | 52.65 | 77.33 | 50.69 | 45.07 | 73.56 | 59.86 |
| LoRA-Switch (origin) | 52.39 | 77.60 | 51.15 | 46.15 | 73.32 | 60.12 |

Table 8: Ablation study on Domain specific LLM ability improvements of LoRA-Switch.

| Methods | ScienceQA | CommonsenseQA | OpenbookQA | Avg |
|---|---|---|---|---|
| Llama2-7B | 53.19 | 47.82 | 45.80 | 48.94 |
| LoRA-Switch (16 expert) | 92.09 | 76.90 | 77.60 | 82.20 |
| LoRA-Switch ($r$=32) | 89.93 | 77.23 | 79.60 | 82.25 |
| LoRA-Switch ($r$=16) | 91.64 | 76.25 | 76.80 | 81.56 |
| LoRA-Switch (Top-1) | 91.68 | 74.77 | 76.80 | 81.08 |
| LoRA-Switch (origin) | 91.39 | 79.03 | 80.40 | 83.60 |

As depicted in Table 7 and Table 8, the experimental results indicate that LoRA-Switch consistently enhances model accuracy across various adapter configurations. When the rank $r$ of the LoRA experts is set to 32 or 16, the model exhibits relatively lower accuracy. This outcome suggests that a lower rank might not provide sufficient model capacity to assimilate a broad spectrum of new information effectively. Furthermore, using Top-1 routing results in a decrease in model accuracy compared to the original Top-2 routing. This decline can be attributed to the fact that, in Top-1 routing, each input token is directed to only one expert, whereas in Top-2 routing, tokens are processed by two experts. Generally, the more experts a token interacts with, the better the expected performance due to enhanced processing capabilities. Lastly, our findings reveal that increasing the number of experts to sixteen per layer paradoxically leads to a decrease in accuracy. This suggests that while

the model's capacity is augmented, the available data for fine-tuning is insufficient to fully optimize the expanded model capabilities. This discrepancy underscores the need for a balanced approach to configuring dynamic adapters, where both the number of experts and the rank of adapters must be carefully calibrated to the available training resources.

## B.2 ABLATION STUDY ON RUNTIME LATENCY

Subsequently, we investigated the effects of different components within LoRA-Switch. Specifically, we evaluated the impact of replacing the SGMM with a simple merge method to assess how such changes affect the system's performance. This part of our study is crucial for understanding the contribution of each component to the overall effectiveness of LoRA-Switch.

Table 9: Ablation study for runtime latency of LoRA-Switch.

| Method | Decoding latency (ms/token) |
|---|---|
| Llama2-7B | 2.4 |
| MoRAL | 8.5 (+254%) |
| MoRAL (With simple merge) | 4.5 (+88%) |
| LoRA-Switch (With simple merge) | 4.2 (+ 75%) |
| LoRA-Switch | 3.1 (+29%) |

As presented in Table 9, replacing the Sparse Generalized Matrix Multiplication (SGMM) in our proposed LoRA-Switch with a simple merge approach results in a substantial increase in decoding latency, rising from 3.1 ms/token to 5.1 ms/token. This represents a 113% increase compared to the original Llama2-7B model, underscoring the significant efficiency gains achieved by SGMM during the decoding phase and the consequent reduction in LoRA-Switch's decoding inference latency.

We further explored the integration of the simple merge technique into the MoRAL method. Here, after determining which experts to activate within each dynamic adapters layer, we first merge these experts into the base model before inference. This method succeeded in reducing the decoding latency to 4.5 ms/token; however, it still registered an 88% increase over the original Llama2-7B. These findings highlight the efficiency of our LoRA-Switch framework, which synergistically optimizes at both the algorithmic and system levels, demonstrating the effectiveness of our novel dynamic adapters architecture.

