# OpenReview forum: "LoRA-Switch: Boosting the Efficiency of Dynamic LLM Adapters via System-Algorithm Co-design"
_ICLR.cc/2025/Conference — Submitted to ICLR 2025_

### Official Review · Reviewer_finz · 2024-11-02

**Soundness:** 3
**Presentation:** 1
**Contribution:** 2
**Rating:** 5
**Confidence:** 3

**Summary:**

This paper presents LoRA-Switch, an architecture for efficient (in particular, low-latency) adapters. The authors first conducted a fine-grained analysis of the decoding latencies of existing adapter architecture like MOLA and MoRAL, and found that they introduce significant latency overhead despite minor impact on parameter size and computational complexity. The proposed architecture, LoRA-Switch, addresses this issue through several design choices: (1) Token-wise, instead of block-wise or layer-wise, adapter routing (2) Reducing CUDA kernel execution overhead through pre-merging activated LoRA adapters into the pretrained model's backbone before decoding (3) The development of a fused CUDA kernel called SGMM for managing activated/inactivated adapters. Experiment results show that LoRA-Switch is comparable with baselines in terms of accuracy, but incurs much lower decoding latency at runtime.

**Strengths:**

`+` Solid analysis of existing adapters and their common performance issue --- High decoding latency overhead. The motivation for the paper is thus very clear even to non-expert audience.

`+` Evaluation results are good --- LoRA-Switch can achieve almost the same level of accuracy (in incremental training & domain-specific fine-tuning) as the baselines, and the decoding latency is much lower. The additional overhead in peak memory is very small, so the performance improvement in latency almost seems like free-lunch to me.

**Weaknesses:**

`-` Even though I get the intuition of some design choices (e.g. How you reduce the number of CUDA kernel calls; Why you design the SGMM kernel), the overall design section is very convoluted and difficult to understand, especially section 3.4. Although I appreciate the authors' efforts in explaining all the details, it would be better to give some more higher-level insights and intuitions for the ease of understanding.

`-` Given the scale of the experiments in evaluation, it is unclear if the architecture would (1) scale to larger pre-trained backbone models, like 70B+ LLMs, and (2) scale to a larger number of LoRA adapters, e.g., a common setup in research/industry is 128 adapters, and a different setup for the router, e.g. top-K with a larger K. I fully understand that it can be very difficult to run experiments with larger pre-trained backbone models or a larger number of LoRA adapters, and this comment is NOT a request for the authors to add additional experiment results --- However, from a research perspective, it would be greatly appreciated if the authors can provide some insights and intuitions on why LoRA-Switch could (or could not) be a general solution to dynamic adapters architecture even when the scale goes up; otherwise, it would seem like that the architecture is a specific engineering solution to a specific scenario (7B models, 8 adapters, and top-2 router).

**Questions:**

Please refer to the "weaknesses" session.

**Details Of Ethics Concerns:**

This paper does not raise any ethics concerns.

---

### Official Review · Reviewer_ubWo · 2024-11-02

**Soundness:** 3
**Presentation:** 3
**Contribution:** 3
**Rating:** 5
**Confidence:** 3

**Summary:**

The paper presents LoRA-Switch, aiming to reduce inference latency in large language models (LLMs) that use dynamic adapters. Dynamic adapters like LoRA and Mixture-of-Experts (MoE) are effective for adapting models to various tasks but often suffer from high latency due to the need for multiple, frequent GPU calls. LoRA-Switch addresses this by implementing a Top-2 token-wise routing mechanism and optimizing the fusion of adapters into the model backbone.

**Strengths:**

+Authors reveal the somewhat counterintuitive fact that while the MoE + LoRA method theoretically does not add significant FLOPS, it leads to a substantial increase in latency in practical applications.

+To address the inference latency overhead due to dynamic adapters, the authors proposed the following techniques:
1. Top-2 Adapter Selection: LoRA-Switch uses a Top-2 router to dynamically select the two most relevant adapters for each token, applying them consistently across all layers, which minimizes GPU calls and reduces latency.
2. Fused Adapter Switching: To further reduce overhead, LoRA-Switch merges selected adapters into the backbone at each token step. The SGMM (Segmented Gather Matrix Multiplication) kernel batches these operations into a single GPU call, enhancing efficiency.
3. Pre-Gated Structure: LoRA-Switch shares routing weights across layers and applies the same adapters throughout, simplifying computation and lowering latency by eliminating per-layer routing decisions.

+The results show that the proposed method matches or outperforms existing dynamic adapter methods in both accuracy and latency. In addition, it reduces decoding latency by over 2.4x compared to alternatives like MoRAL and PESC.

**Weaknesses:**

-If the reviewer’s understanding is correct, the authors essentially remove all gatings except for the first linear layer, and apply the result across all layers, without providing insights into why this approach would not affect the model's performance.

-Similarly, the design choice of computing the MoE gating only once and fusing LoRA across all layers seems unrelated to LoRA itself, as conventional MoE can also be viewed as fusing selected experts into a single matrix at each layer. The authors do not explain the necessity of incorporating LoRA into this design.

-The strong coupling between the algorithmic and system-level optimizations makes the method unsuitable for application to an off-the-shelf MoE + LoRA model.

More specifically:

A potential limitation arises because the same top-2 adapters are applied across all layers for each token. In a deep model, different layers often extract distinct features, which could benefit from layer-specific adapter selection. By using the same adapter selection across layers, LoRA-Switch sacrifices some granularity, potentially missing layer-specific optimizations for certain tokens.

The proposed method may imply a potential trade-off between efficiency and layer-specific adaptability. While LoRA-Switch achieves faster processing, it could benefit from mechanisms that adapt adapter importance at each layer, potentially enhancing layer-level performance without reintroducing full per-layer routing.

**Questions:**

Please see the above review. In addition, the reviewer is unclear on the exact workflow described by the two MoRAL items in Table 9. Does it indicate that the mechanism of performing gating only once at the first LoRA layer results in a 166% increase in inference speed? If so, the CUDA-level optimization appears less critical compared to the algorithm-level optimization.

---

### Official Review · Reviewer_SDHu · 2024-11-02

**Soundness:** 3
**Presentation:** 3
**Contribution:** 3
**Rating:** 5
**Confidence:** 3

**Summary:**

The authors devise a scheme to mitigate the latency overheads of MoE-based LoRA adapters. They do this by avoiding block/layer-wise adapter routing but rather adopt token-wise adapter routing - i.e. making routing decisions and selecting adapters once, and only once per token. The activated LoRA adapters are pre-merged into the backbone for each token's decoding phase. This reduces dependencies / kernal calls and significantly reduces latency. Results compare to fine-tuning using LoRA, MoRAL, MOLA, PESC and full fine-tuning.

**Strengths:**

The paper presents and evaluates a relatively simple way to avoid the additional latency incurred by other MoE adapter schemes. Results indicate that the scheme works well, both in terms of performance and in reducing latency.

**Weaknesses:**

Would this approach not have quite serious implications for batching?

**Questions:**

Q1: What are the implications for batching, i.e. what if tokens within a batch require different adapters? Is you evaluation for a batch size = 1?

---

### Official Review · Reviewer_HVpG · 2024-11-04

**Soundness:** 2
**Presentation:** 2
**Contribution:** 2
**Rating:** 3
**Confidence:** 4

**Summary:**

This paper analyzes how dynamic adapter methods impact inference latency and
proposes a token-wise routing approach to speed up inference.
While existing dynamic adapter methods improve the capabilities of
pre-trained models, they increase the inference latency by 3.5-10.5x.
The authors attribute this slowdown to the increased number of CUDA kernel calls
needed to support dynamic adapters.
To address this issue, the paper proposes LoRA-Switch, a new dynamic adapter method
that uses token-wise routing and fuses adapter weights into the backbone of the model.
This approach significantly reduce the dynamic adapter overhead, resulting in a slowdown
of only 0.29x, which achieving comparable accuracy to existing dynamic adapter methods.
The paper attributes this speedup to the SGMM kernel which efficiently switches adapters and fuses
them to the backbone in a single CUDA kernel call.

**Strengths:**

1. Efficient kernel which demonstrates strong latency improvements over existing dynamic adapter implementations.
2. LoRA-Switch's accuracy is close to state-of-the-art dynamic adapter methods despite using token-wise routing.

**Weaknesses:**

1. Lacks discussion or evaluation of batched inference and inference throughput, which limits the impact of the work. The paper's evaluation focuses on decoding latency for batch size 1, which is applicable to latency-critical applications that serve few requests (e.g., edge deployments). Demonstrating that LoRA-Switch provides better throughput for larger batch sizes and diverse request workloads would make this approach more compelling for general-purpose and large-scale LLM deployments.
2. The paper attributes latency improvements to reducing the number of CUDA kernel calls. Kernel fusion – which is used in Jax, torch.compile, and by other ML compilers – similarly reduces the number of CUDA kernel calls, which may be as low as 1 call per forward pass (or training step). If the reduced number of CUDA calls is the primary reason for the improvement in decoding latency, then the baseline implementations are likely sub-optimal for batch-size 1 decoding. Moreover, the baseline implementations could be optimized for throughput (i.e., training or batched inference) in which case the overheads of calling a kernel may be minimal due to a larger amount of work done per kernel invocation. Because throughput-focused implementations often come at a cost of increased latency, I am concerned that the reported performance improvements might be disproportionately large. An improved evaluation could apply kernel fusion to the other dynamic adapter implementations (or explain why this is not possible), and compare the performance improvement of LoRA-Switch relative to the baselines with kernel fusion.

**Questions:**

1. How much of the SGMM kernel's speedup is due fewer dispatches vs. fusing the adapters with the backbone model?
2. How does the decoding speed of LoRA-Switch compare to implementations for the baselines that use fused kernels?
3. How does the throughput of LoRA-Switch compare for (i) large batch sizes that select the same adapter, and (ii) larger batch sizes selecting diverse expert adapters?
4. The abstract claims that current dynamic adapter methods increase decoding latency by over $2.5\times$ while the results claim a slowdown of over 250% (3.5x). What is the true measured slowdown?
5. Is it possible to include decoding profiles for the various methods (e.g., using [NVIDIA Visual Profiler](https://developer.nvidia.com/nvidia-visual-profiler) or [PyTorch Profiler](https://pytorch.org/tutorials/recipes/recipes/profiler_recipe.html)? This could help visualize how much overhead is due to kernel invocation overheads and GPU under-utilization.

---

### Meta-Review · Area_Chair_9hJn · 2024-12-21

**Metareview:**

**Summary:** The paper proposes LoRA-Switch, a system-algorithm co-design to reduce inference latency for dynamic adapters in LLMs. It introduces token-wise routing for adapter selection, combined with an optimized CUDA kernel to merge selected adapters into the backbone, achieving over 2.4× decoding latency reduction compared to existing dynamic adapters.

**Strength:**

1. Practical system-level optimizations through fused CUDA kernel operations for dynamic adapter merging.

2. Solid motivation and analysis of latency bottlenecks in existing dynamic adapter methods.

**Weakness:**
1. This work lacks discussion and evaluation of batched inference and inference throughput.

2. Potentially unfair comparisons arise due to the lack of kernel fusion applied to the baseline methods.

3. The paper writing lacks clarity in presenting design decisions and convoluted explanations in key sections, such as Section 3.4.

4. The insufficient evaluation on diverse settings, such as larger batch sizes and larger transformer architectures, limits the generality and applicability of the proposed method.

**Reasons for the decision:**

Since the evaluations are limited to small-scale setups and fail to convincingly demonstrate the method's scalability across batched inference and larger-scale models, coupled with unclear explanations of critical design choices that make the contributions less accessible, I recommend rejection.

**Additional Comments On Reviewer Discussion:**

The authors did not provide a rebuttal, and all reviewers leaned toward rejecting this paper. I concur with this decision.

---

### Decision · Program_Chairs · 2025-01-22

Reject